# Optical Fiber Temperature and Humidity Dual Parameter Sensing Based on Fiber Bragg Gratings and Porous Film

**DOI:** 10.3390/s23177587

**Published:** 2023-09-01

**Authors:** Jiankun Peng, Jianren Zhou, Chengli Sun, Qingping Liu

**Affiliations:** School of Information Engineering, Nanchang Hangkong University, Nanchang 330063, China

**Keywords:** optical fiber sensing, fiber Bragg gratings, porous films, humidity

## Abstract

A porous anodic alumina film is proposed to construct an optical fiber temperature and humidity sensor. In the sensor structure, a fiber Bragg grating is used to detect the environment temperature, and the porous film is used to detect the environment humidity. The proposed porous anodic alumina film was fabricated by anodic oxidation reaction, and it is suitable for the use of humidity detection due to its porous structure. Experimental results show the temperature sensitivity of the proposed sensor was 10.4 pm/°C and the humidity sensitivity of the proposed sensor was 185 pm/%RH.

## 1. Introduction

Optical fiber sensors are widely used in the industry on account of its excellent performance on electromagnetic resistance, remote sensing, and safety. Optical fiber temperature sensors and optical fiber humidity sensors have important sensing applications in the environment monitoring of the industry. There are various sensing configurations of the optical fiber temperature sensor and the optical fiber humidity sensor. For the optical fiber temperature sensor, four main sensing structures have been proposed by previous researchers, which are fiber Bragg gratings (FBG) [1,2], Fabry−Perot interferometers (FPI) [3,4], and surface plasmon resonance (SPR) [5,6] and tapered fibers [7,8]. Among these fiber temperature-sensing structures, the FBG is very suitable for the use of temperature detection due to its linear response to temperature, the capability of distributed detection, and simple manufacturing processes. For the optical fiber humidity sensor, the humidity-sensing structure has to contain a humidity-sensitive film which is used to convert the humidity variation into the light signal change. In general, polymer and ceramic are the two main categories of humidity-sensitive materials. Polymer humidity-sensitive materials contain chitosan [9], agarose [10], nafion [11], polyvinyl alcohol [12], polyimide [13], and so on. The polymer humidity-sensitive film will expand when water molecules are absorbed into the film, and it is usually used in the high-humidity environment. Ceramic humidity-sensitive materials contain SiO_2_ [14], SnO_2_ [15], ZnO [16], TiO_2_ [17], Al_2_O_3_ [18], and so on. The optical path difference of the ceramic humidity-sensitive film will change as water molecules are absorbed into the film, and it still has good performance within the ultra-low humidity environment [19]. In most environments, it is needed to monitor temperature and humidity simultaneously; hence, dual-parameter-sensing structures for the optical fiber temperature and humidity [20,21,22,23,24,25] have been proposed. SPR structures which are based on the no-core optical fiber [20] and the double D-shaped optical fiber [21] are very difficult to fabricate. The Fabry−Perot structure based on the photonic crystal fiber [22] is hard to reproduce sensors with the same parameters. Long-period optical fiber gratings [23] and optical fiber Bragg gratings [24] are suitable for cascade connection and robustness. Mesoporous dielectric films [23,25] are suitable for the use of low-environment-humidity detection. Therefore, a sensor having a structure combining optical fiber Bragg gratings with porous dielectric films can be used as a temperature and humidity sensor within low-humidity environments.

In this paper, an optical fiber temperature and humidity sensor is presented. The proposed optical fiber sensor consists of a fiber Bragg grating and a porous anodic alumina film. The FBG within the sensor structure is used to detect the environment temperature. The proposed porous anodic alumina film was fabricated by anodic oxidation reaction. Due to its capability of water adsorption, it was used to detect the environment humidity. The fabrication processes of the sensor probe and its demodulation system are introduced in detail. At the end, we analyzed the experimental results of the prepared sensor on the temperature detection and the humidity detection, and then we concluded our findings.

## 2. Theory of Sensor Detection

### 2.1. Temperature Sensing Theory

The fiber Bragg grating is a kind of optical passive device, which has the capacity of reflecting light with a Bragg wavelength. For the FBG structure, if the effective refractive index and the refractive index period are $n_{eff}$ and Λ, respectively, then the Bragg wavelength can be calculated by Formula (1) [26]:(1)$$\lambda_{B}=2n_{eff}\Lambda$$

When the environment temperature is changed, the effective refractive index and the refractive index period of the FBG will change with the temperature; therefore, the Bragg wavelength will be shifted. The relationship between the Bragg wavelength and the temperature can be expressed as the derivative of $\lambda_{B}$ with respect to *T*. Herein, *T* is the temperature. In Equation (2), the differential of $\lambda_{B}$ is calculated based on Formula (1):(2)$$d\lambda_{B}=2n_{eff}d\Lambda +2\Lambda dn_{eff}=2n_{eff}\frac{\partial \Lambda }{\partial T}dT+2\Lambda \left(\frac{\partial n_{eff}}{\partial T}dT+\frac{\partial n_{eff}}{\partial \lambda }d\lambda \right)$$

If the thermal expansion coefficient and the thermo-optical coefficient are α and ζ, respectively, the refractive index and the refractive index period can be expressed as Equations (3) and (4), respectively. In Equations (3) and (4), $\Lambda_{0}$ is the initial refractive index period, $n_{0}$ is the initial refractive index, and $T_{0}$ is the initial temperature.
(3)$$\Lambda =\Lambda_{0}\left[1+\alpha \left(T-T_{0}\right)\right]$$
(4)$$n=n_{0}+\zeta n_{0}\left(T-T_{0}\right)$$

From Equations (3) and (4), it can be calculated that the differential of Λ is $d\Lambda =\alpha \Lambda_{0}dT$ and the differential of *n* is $dn=\zeta n_{0}dT$. The initial Bragg wavelength is $\lambda_{0}=2n_{eff}\Lambda_{0}$. Substituting the parameters into Equation (2), the differential of $\lambda_{B}$ can be deduced as Equation (5):(5)$$d\lambda_{B}=\alpha \lambda_{0}dT+\zeta \lambda_{0}dT+\frac{\lambda_{0}}{n_{eff}}\frac{\partial n_{eff}}{\partial \lambda }d\lambda$$

Due to $\frac{\lambda_{0}}{n_{eff}}\frac{\partial n_{eff}}{\partial \lambda }\ll 1$, i.e., the impact of this part is extremely minimal, it can be ignored. For the FBG, the thermal expansion coefficient and the thermo-optical coefficient are constant. Hence, the coefficient of the Bragg wavelength and temperature can be calculated as Equation (6). In Equation (6), $K_{B,T}$ is the temperature sensitivity coefficient of the FBG, and $\lambda_{0}$ is the initial central wavelength value of the FBG.
(6)$$K_{B,T}=\frac{d\lambda_{B}}{dT}=\lambda_{0}\left(\alpha +\zeta \right)$$

For the germanium-doped silica fiber, $\alpha =5.5\times 10^{-7}$/°C, and $\zeta =6.4\times 10^{-6}$/°C. In addition, when the initial Bragg wavelength is 1550 nm, it can be deduced that the temperature sensitivity coefficient of the FBG is 0.0109 nm/°C. Therefore, the central wavelength shift of the FBG has a linear response to temperature; hence, the FBG is suitable for temperature sensing.

### 2.2. Humidity Sensing Theory

The anodized alumina film is a porous thin film, and it has the capability of adsorbing water molecules. On the surface of aluminum oxide, the first layer of adsorptive water molecules is bonded by the chemisorption, and the following layers of adsorptive water molecules are bonded by the physisorption. Due to the fact that the anodized alumina film has a porous structure, it has a great superficial area to adsorb water molecules. In addition, the adsorptive water molecules are stored in the columnar porous of the film due to the condensation effect. As a result, in the humid environment, the anodized alumina film contains three components which are the film material, condensed water, and air. For the three-component system of the thin film, the Bruggeman medium model is an effective method to calculate the effective refractive index, which is expressed as follows [27]:(7)$$\left(f-V\right)\frac{1-n_{e}^{2}}{1+2n_{e}^{2}}+\left(1-f\right)\frac{n_{Al_{2}O_{3}}^{2}-n_{e}^{2}}{n_{Al_{2}O_{3}}^{2}+2n_{e}^{2}}+V\frac{n_{H_{2}O}^{2}-n_{e}^{2}}{n_{H_{2}O}^{2}+2n_{e}^{2}}=0$$

In Equation (7), $n_{Al_{2}O_{3}}$ and $n_{H_{2}O}$ are the refractive indices of the anodized alumina and water, respectively, $n_{e}$ is the effective refractive index of the thin film, and *f* and *V* are the volume fractions of the pores and the water, respectively. When the water volume fraction is changed, the effective refractive index of the thin film will change. Therefore, due to the water adsorption and the condensation, the effective refractive index of the anodized alumina film will change with the environment humidity.

The anodized alumina film has two reflection interfaces, and it constitutes a Fabry−Perot structure. When the light is transmitted into the film, an equal-thickness interference will occur within the film. Due to the fact that the high-order reflective light is extremely weak, the light interference of the thin film can be simplified as a dual-beam interference. The total reflective light intensity of the dual-beam interference can be expressed as Formula (8) [28]:(8)$$I=I_{1}+I_{2}+2\sqrt{I_{1}I_{2}}\cos\left(\frac{2\pi }{\lambda }\Delta \right)$$

In Formula (8), *I*_1_ and *I*_2_ are the light intensities of the two reflective light which are reflected from the two interfaces of the thin film, λ is the wavelength of the incident light, and Δ is the optical path difference of the two reflective light. If the thin film thickness is L, then $\Delta =2n_{e}L+\lambda /2$. Thus, the total reflective light intensity is changed with the variation of *n_e_*. Furthermore, the *n_e_* of the anodized alumina film is changed with the environment humidity. In brief, the interference spectrum of the anodized alumina thin film is influenced by the environment humidity on account of the *n_e_* of the thin film is changed with the variation of the environment humidity. Therefore, the porous anodized alumina thin film can be used as a humidity-sensing material.

## 3. Sensor Fabrication and Demodulation

The porous thin film was manufactured by anodic oxidation reaction. In the anodic oxidation reaction, the aluminum was used as the anode. When a voltage was applied, anions in the electrolyte converged towards the aluminum anode, where the aluminum lost electrons and became trivalent aluminum ions. Aluminum ions which were dissolved in the solution might combine with anions (i.e., oxygen ions) to form compounds. The result of the reaction is related to the electrolyte of the solution, and the current, the voltage, the temperature and the period of the oxidation process. When the electrolyte of the solution was sulfuric acid, the porous alumina film was colorless and transparent. The formation process of the porous alumina film had six steps, which are depicted as follows. (1) When a voltage was applied, the nonporous alumina film started to form on the surface of the aluminum sheet. (2) As the nonporous alumina film was growing thick, the surface of the film was uneven due to the film expansion. (3) The pores were gradually generated on account of the uneven surface and the uneven electric field. (4) When the electric field was gradually equilibrium since the continuous formation and dissolution of the anodic alumina films, the random pores became regular and ordered. (5) When the alumina films were removed by corrosion, there were regular and orderly grooves left on the aluminum substrate. (6) In the second anodization process, the required porous anodic alumina films could be generated on the aluminum substrate with the same voltage and parameters, since the existence of regular and orderly grooves. The surface Scanning Electron Microscope (SEM) image of the manufactured porous anodic alumina film is shown in Figure 1.

Figure 1 shows the porous structure of the thin film surface, and this surface is toward the air. It can be seen from Figure 1 that a large number of columnar capillary pores were distributed within the thin film. These pores increased the specific surface area of the thin film and improved the water molecule adsorption ability of the thin film. In addition, these pores had to be open, and then the water molecules were adsorbed into the thin film. Furthermore, big pores were better than small ones on account of their bigger specific surface area.

In the sensor structure fabrication process, in order to use the porous anodic alumina films, it had to remove the aluminum substrate with a saturated copper chloride solution by replacement reaction. When the FBG was inscribed on the single-mode optical fiber, then a ceramic ferrule was connected to the optical fiber end with an ultraviolet adhesive after removing the coating layer of the fiber end. Next, the porous anodic alumina film could be adhered to the end face of the ceramic ferrule, and the pores of the porous anodic alumina film had to face the air. The Bragg wavelength of the fabricated sensor was 1550 nm, and the thickness of the porous film was 50 μm. The configuration of the fabricated fiber temperature and humidity sensor is shown in Figure 2. In environmental monitoring, there was an armored protection around the sensor probe to prevent the destruction of the sensor.

The reflected spectrum of the proposed fiber temperature and humidity sensor can be detected by using an optical fiber C band demodulation system, since the interference spectrum and the Bragg wavelength were located in the C band. As shown in Figure 3, the C band demodulation system consisted of a C band light source, an optical spectrometer (YOKOGAWA, AQ6370B; wavelength resolution: 0.02 nm), and an optical single-mode fiber coupler. The single-mode fiber coupler was used to connect the proposed fiber temperature and humidity sensor with the infrared light source and the optic spectrometer. The computer was used to process detected data. In the testing experiment, different saturated salt solutions were used to create different humid environments, and the water bath kept the temperature stable. In the detection, the change of temperature or humidity was converted into the wavelength shift of the sensor spectrum.

## 4. Experimental Results

The reflective spectrum of the fabricated temperature and humidity sensor is shown in Figure 4. The wavelength of the infrared light source ranged from 1520 nm to 1620 nm. In Figure 4, it can be clearly noticed that there was a reflection peak of FBG at 1550 nm and an interference spectrum of the porous anodic alumina film.

At first, when the humidity was maintained at an 84% RH, the temperature-sensing performance of the sensor was investigated. During the temperature experiment, in order to stabilize the interference spectrum of the sensor, the sensor was put into a stable humidity environment which was created with a saturated potassium chloride solution. Subsequently, an airtight bottle (the sensor probe and the saturated salt solution were within it) was put into a blast drying oven (9055A; Shanzhi Instrument Equipment Co., Ltd., Shanghai, China) to conduct the temperature test. When the temperature value set was reached, the environment temperature was maintained for 10 min. The reflective spectra of the FBG at different temperatures are shown in Figure 5a. As shown in Figure 5a, the Bragg wavelength of the temperature and humidity sensor showed a red-shift with the rise of temperature. For the temperatures of 30 °C, 40 °C, 50 °C, 60 °C, 70 °C, and 80 °C, the peak wavelengths of the FBG were 1549.78 nm, 1549.88 nm, 1549.98 nm, 1550.08 nm, 1550.18 nm, and 1550.3 nm, respectively. The relationship of the peak wavelength and its corresponding temperature is plotted and fitted in Figure 5b. The fitted line shows the peak wavelength was linearly responsive to the temperature, and its linearity was 0.99949. The temperature detection experimental results reveal that the prepared optical temperature and humidity sensor had a temperature sensitivity of 10.4 pm/°C.

After the temperature experiment, the humidity-sensing performance of the sensor was investigated with the temperature maintained at 25 °C and the saturated salt solutions. Different humidity environments within the testing bottle were generated by the saturated salt solutions which were prepared with lithium chloride (11.3% RH, 25 °C), potassium acetate (22.51% RH, 25 °C), potassium carbonate (43.16% RH, 25 °C), sodium bromide (57.57% RH, 25 °C), and potassium chloride (84.34% RH, 25 °C) separately. The reflective spectra of the temperature and humidity sensor in different humidity environments are shown in Figure 6a. As shown in Figure 6a, the interference spectrum of the porous anodic alumina film showed a red-shift with the increase of humidity. In other words, the interference spectrum of the sensor moved to a longer wavelength, when the environment humidity was increased. The troughs of the interference spectrum was suitable for wavelength extraction. As shown in Figure 6a, the trough near 1550 nm was disturbed by the FBG peak, and the trough near 1590 nm drifted out of the detection area of the optical spectrometer if the relative humidity was high than an 80% RH; however, the trough near 1570 nm did not have the above interfering factors. In order to clearly show the wavelength shift, the spectra from 1565 nm to 1590 nm are magnified, which are put in Figure 6a. In the magnified spectra, it can be clearly observed that the wavelength shift of the characteristic valley was 13 nm with the humidity increase from 11% RH to 84% RH. When the environment humidity values were 11% RH, 22% RH, 43% RH, 57% RH, and 84% RH, the wavelengths of the characteristic valley were 1570.8 nm, 1573.7 nm, 1575.7 nm, 1578.1 nm, and 1584.3 nm, respectively. The relationship of the characteristic wavelength and its corresponding humidity is plotted and fitted in Figure 6b. The humidity detection experimental results reveal that the prepared optical temperature and humidity sensor had a humidity sensitivity of 185 pm/%RH, and it is linearly responsive to the humidity with a linearity of 0.98787.

The humidity-sensing performance of the sensor compared with previous publications is shown in Table 1. From Table 1, it can be seen that ceramic materials can be used as a humidity-sensitive film of ultra-low humidity detection. The anodized alumina film is easy to control its porosity and thickness, which makes it suitable for the ultra-low humidity detection. In ultra-low humidity environments, the humidity detection is strongly influenced by temperature. Thus, the sensor configuration within this paper will contribute to investigating the relationship between humidity and temperature at ultra-low humidity environments.

## 5. Conclusions

In conclusion, the current work presents an optical fiber temperature and humidity sensor with a fiber Bragg grating and a porous anodic alumina film. The FBG is used to detect the environment temperature, and the porous anodic alumina film is used to detect the environment humidity on account of its capability of water adsorption. The proposed porous anodic alumina film was fabricated by anodic oxidation reaction. Temperature detection results show the temperature sensitivity of the sensor was 10.4 pm/°C, and humidity detection results show the humidity sensitivity of the sensor was 185 pm/%RH.

## Figures and Tables

**Figure 1 sensors-23-07587-f001:**
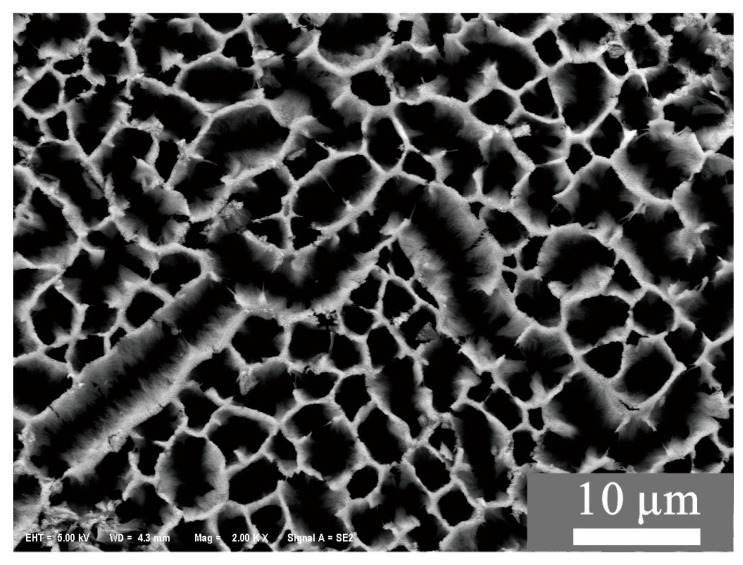
SEM image of the porous alumina film (surface, Mag = 2.00 K×).

**Figure 2 sensors-23-07587-f002:**
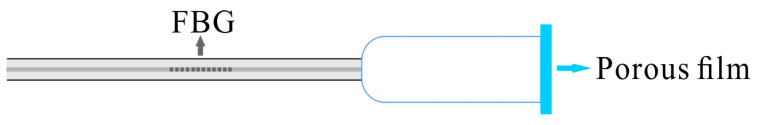
Configuration of the temperature and humidity sensor structure.

**Figure 3 sensors-23-07587-f003:**
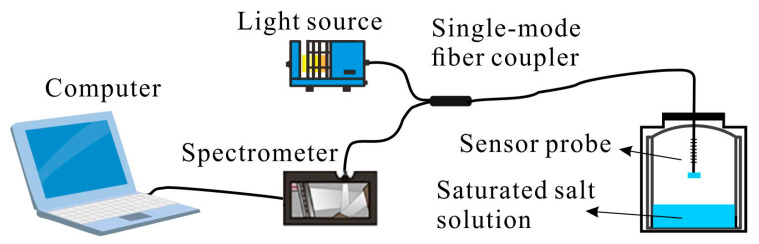
Configuration of the optical fiber C band demodulation system.

**Figure 4 sensors-23-07587-f004:**
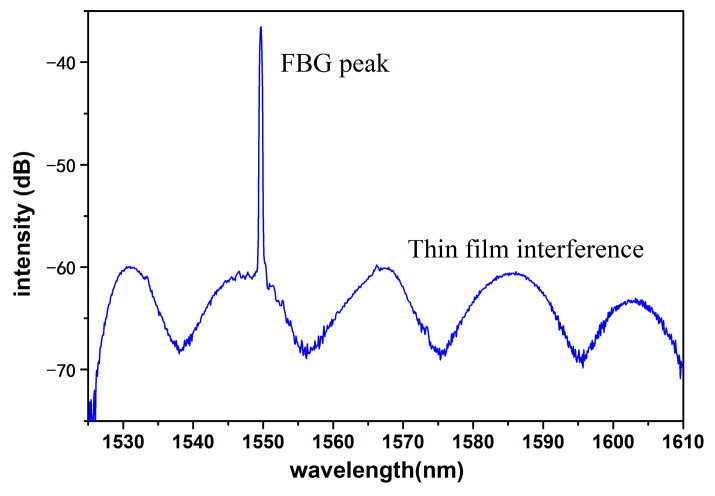
Reflective spectrum of the temperature and humidity sensor.

**Figure 5 sensors-23-07587-f005:**
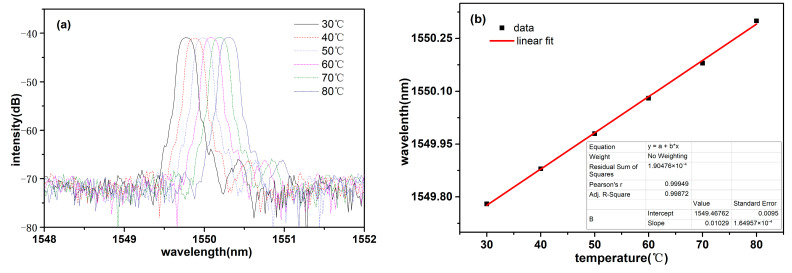
(**a**) Reflective peaks of the FBG at different temperatures; (**b**) response of the Bragg wavelength to temperature.

**Figure 6 sensors-23-07587-f006:**
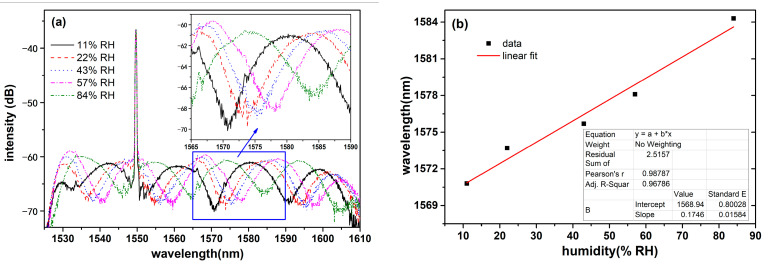
(**a**) Interference spectra of the porous film at different humidity values; (**b**) response of the characteristic wavelength to humidity.

**Table 1 sensors-23-07587-t001:** Performance comparison of the current work with previous publications.

Prior Art	Material	Measurement Range (%RH)	Sensitivity
[9]	Chitosan	40–92	7.15 nm/% RH
[10]	Agarose	10–95	0.0024 nm·%^−1^
[11]	Nafion	30–85	3.78 nm/%RH
[12]	Polyvinyl alcohol	>70	−0.4573 dB/%RH
[13]	Polyimide	23.8–83.4	1.832 pm/%RH
[14]	SiO_2_	0.06–70	0.3 nm/%RH
[15]	SnO_2_	20–90	1.9 nm/%RH
[16]	ZnO	20–95	25.2 mV/%RH
[17]	TiO_2_	1.8–74.7	0.43 nm/%RH
[18]	Al_2_O_3_	3–98 (and 180–1000 ppm)	/
Reported work	Al_2_O_3_	11.3–84.3	185 pm/%RH

## Data Availability

Data sharing is not applicable to this article.

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
