# Peer review of "Optical Fiber Temperature and Humidity Dual Parameter Sensing Based on Fiber Bragg Gratings and Porous Film"

_sensors, 2023, doi:10.3390/s23177587_

Round 1

Reviewer 1 Report

This manuscript reported an optical fiber temperature and humidity dual parameter sensing based on fiber Bragg gratings and porous film. Experimental results show that the temperature sensitivity of the proposed sensor is 10.4 pm/°C and the humidity sensitivity of the proposed sensor is 185 pm/%RH. However, several questions should be explained before considering for publication.

1. The statement and explanation of Fig.1 should be given in the manuscript. In addition, how to estimate the performance of the porous alumina film from SEM image.

2. Form Fig.5 and Fig.6, the author selected the spectrums from 1565 nm to 1590 nm for analyzing the humidity, why? Please give the reason.

3. Please give the full name of SEM.

4. The initial temperature and humidity should be stated before experiments.

5. Please explain how to judge the robustness of the sensor.  

Author Response

This manuscript reported an optical fiber temperature and humidity dual parameter sensing based on fiber Bragg gratings and porous film. Experimental results show that the temperature sensitivity of the proposed sensor is 10.4 pm/°C and the humidity sensitivity of the proposed sensor is 185 pm/%RH. However, several questions should be explained before considering for publication.

Response: We thank the Reviewer’s valuable and important comments on our manuscript. According to the Reviewer’s comments and kind suggestions, we have revised the manuscript. We have promoted the language of the manuscript.

  1. The statement and explanation of Fig.1 should be given in the manuscript. In addition, how to estimate the performance of the porous alumina film from SEM image.

   Response: Thanks for the suggestion. We have added the statement and explanation of figure 1. The added content is as follows.

Figure 1 shows the porous structure of the thin film surface, and this surface is toward to the air. It can be seen from the figure 1 that a large number of columnar capillary pores are distributed within the thin film. These pores increase the specific surface area of the thin film, and improve the water molecule adsorption ability of the thin film. And these pores have to be open, then the water molecules can be adsorbed into the thin film. Furthermore, big pores are better than small ones on account of the specific surface area of the big pores are higher.

  1. Form Fig.5 and Fig.6, the author selected the spectrums from 1565 nm to 1590 nm for analyzing the humidity, why? Please give the reason.

Response: Thank you for the comment. We have added the reason into the manuscript as follows.

The troughs of the interference spectrum is suitable for wavelength extraction. As show in Figure 6 (a), the trough near 1550 nm is disturbed by the FBG peak, and the trough near 1590 nm will drift out of the detection area of the optical spectrometer if the relative humidity is high than 80%RH, however, the trough near 1570 nm do not have the above interfering factors.

  1. Please give the full name of SEM.

Response: The full name of SEM (Scanning Electron Microscope) is added into the manuscript.

  1. The initial temperature and humidity should be stated before experiments.

Response: Thanks for the suggestion. The relevant content is revised as follows.

At the first, when the humidity is maintained at 84%RH, the temperature sensing performance of the sensor is investigated.

After the temperature experiment, when the temperature is maintained at 25°C, the humidity sensing performance of the sensor is investigated with the saturated salt solutions.

  1. Please explain how to judge the robustnessof the sensor.  

Response: Thanks for the comment. In order to judge the sensor robustness, it need to monitor its long term sensing stability. In environmental monitoring, there have an armored protection around the sensor probe to prevent the destruction of the sensor.

Reviewer 2 Report

In this paper (sensors-2576240), the authors proposed an optical fiber temperature and humidity dual parameter sensing based on fiber Bragg gratings and porous film. The topic is interesting for the target readers and the results are basically acceptable. But there are some problems in the writing, presentation, and discussion of the results.

1.        “i.e. organic films and dielectric films”. The classification of materials is somewhat confusing, and organic materials (films) can also be considered dielectric materials (films). The opposite of organic materials should be inorganic materials. In addition, there are carbon, two-dimensional materials, clay materials (attapulgite, halloysite nanotubes, sepiolite fibers), and composite materials as humidity sensing materials. Suggest reclassification and highlight the advantages of inorganic ceramic materials (refer to Sensors Actuators, B Chem. 317 (2020) 128204).

2.        “SiO2 [14], SnO2 [15], ZnO [16], TiO2 [17], Al2O3 [18]”. They can be considered as an inorganic or ceramic materials.

3.        “different saturated salt solutions”. Specific salt and relative humidity (RH) values need to be provided.

4.        Figure 5 and 6: The image serial number “(a), (b)” needs to be embedded.

5.        From the results (Figure 5b and 6b), the temperature response and humidity response are very close. How to identify temperature and humidity?

6.        Response/recovery times are important for sensors. Can you obtain the response and recovery times of the sensor?

7.        What are the advantages of this work compared to the current progress? Are there any advantages in sensor performance?

8.        Check the format of the references.

9.        Check the format. Figure instead of Fig.

 Minor editing of English language required.

Author Response

In this paper (sensors-2576240), the authors proposed an optical fiber temperature and humidity dual parameter sensing based on fiber Bragg gratings and porous film. The topic is interesting for the target readers and the results are basically acceptable. But there are some problems in the writing, presentation, and discussion of the results.

Response: We thank the Reviewer’s valuable and important comments on our manuscript. According to the Reviewer’s comments and kind suggestions, we have revised the manuscript. We have promoted the language of the manuscript.

  1. “i.e. organic films and dielectric films”. The classification of materials is somewhat confusing, and organic materials (films) can also be considered dielectric materials (films). The opposite of organic materials should be inorganic materials. In addition, there are carbon, two-dimensional materials, clay materials (attapulgite, halloysite nanotubes, sepiolite fibers), and composite materials as humidity sensing materials. Suggest reclassification and highlight the advantages of inorganic ceramic materials (refer to Sensors Actuators, B Chem. 317 (2020) 128204).

Response: Thanks for the suggestion. The classification of materials is changed as polymer and ceramic. The revised relevant content is as follows.

In generally, polymer and ceramic are the two main categories of humidity sensitive materials. The polymer humidity sensitive materials contain Chitosan [9], Agarose [10], Nafion [11], Polyvinyl alcohol [12], Polyimide [13] and so on. The polymer humidity sensitive film will expand when water molecules are absorbed into the film, and it is usually used in the high humidity environment. The ceramic humidity sensitive materials contain SiO2 [14], SnO2 [15], ZnO [16], TiO2 [17], Al2O3 [18] and so on. The optical path difference of the ceramic humidity sensitive film will change as water molecules are absorbed into the film, and it still has good performance within the ultra-low humidity environment [19].

  1. “SiO2[14], SnO2 [15], ZnO [16], TiO2 [17], Al2O3 [18]”. They can be considered as an inorganic or ceramic materials.

Response: Thank you for the suggestion. The classification of this materials is changed as “The ceramic humidity sensitive materials”

  1. “different saturated salt solutions”. Specific salt and relative humidity (RH) values need to be provided.

 Response: Thanks for the suggestion. The humidity (RH) values is added into the manuscript. The revised relevant content is as follows.

the saturated salt solutions which are prepared with lithium chloride (11.3%RH, 25°C), potassium acetate (22.51%RH, 25°C), potassium carbonate (43.16%RH, 25°C), sodium bromide (57.57%RH, 25°C) and potassium chloride (84.34%RH, 25°C) respectively.

  1. Figure 5 and 6: The image serial number “(a), (b)” needs to be embedded.

Response: Thanks for the suggestion. The image serial number “(a), (b)” have been embedded into the figure.

  1. From the results (Figure 5b and 6b), the temperature response and humidity response are very close. How to identify temperature and humidity?

Response: Thanks for the comment. The FBG peak (around 1550 nm) is used to identify temperature, and the FBG is insensitive to the humidity. The thin film interference trough (around 1575 nm) is used to identify humidity. The peak shifting range and the trough shifting range are not crossed, therefore, the peak and the trough can identify temperature and humidity respectively.

  1. Response/recovery times are important for sensors. Can you obtain the response and recovery times of the sensor?

Response: Thank you for the comment. For now, we have not tested the response and recovery times of the sensor yet, and it will be studied in the further experiments. Additionally, in a previous research (Sensors & Actuators: B. Chemical 329 (2021) 128908), when the humidity level is 20%-50%RH, the response and recovery time of the anodic aluminium oxide film is 27 s and 40 s respectively.

  1. What are the advantages of this work compared to the current progress? Are there any advantages in sensor performance?

Response: Thanks for the comment. At present, seeking novel humidity sensing materials and synthesizing higher sensitive materials are the main approach to improve the humidity sensor performance. But, in the ultra-low humidity environment, the humidity detection is strongly influenced by temperature. Thus, the sensor configuration within this manuscript will contribute to investigate the relationship between humidity and temperature at low humidity environment.

  1. Check the format of the references.

Response: Thanks for the suggestion. And the reference format has been adjusted.

  1. Check the format. Figure instead of Fig.

Response: Thanks for the suggestion. In the manuscript, all the Fig. are replaced by Figure.

Reviewer 3 Report

The study introduces an optical fiber temperature and humidity sensor utilizing a porous anodic alumina film. The sensor design incorporates a fiber Bragg grating for temperature detection and a porous film for humidity sensing. The experimental results presented in this study demonstrate the reflective spectrum characteristics of the fabricated temperature and humidity sensor.

It can be published once the following issues are addressed.

1. As shown in figure 4, what is the effect of parametric variations on the spectrum?

2. What do you think of this sensor's performance in the O-band?

3. Why was the wavelength range of 1520 nm to 1620 nm chosen for the infrared light source? Is this range relevant to the sensor's application?

4. How was the temperature experiment carried out? Were the temperature conditions accurately controlled and stable during the experiment?

5. Can you elaborate on why the porous film was placed in a stable humidity environment during the temperature experiment? How does humidity stability affect temperature measurements?

6. As shown in Fig. 5, are the reported Bragg wavelength shifts consistent with theoretical expectations for the given temperature range?

7. How confident are you in the accuracy of the 13 nm wavelength shift observed in the characteristic valley? What factors contribute to measurement variability?

8. Were there any comparative analyses of existing sensors or benchmarks in terms of accuracy, sensitivity, or other performance metrics? Please provide a table for comparisons.

Thanks!

The sentence structure could be improved, but the overall content is acceptable.

Author Response

The study introduces an optical fiber temperature and humidity sensor utilizing a porous anodic alumina film. The sensor design incorporates a fiber Bragg grating for temperature detection and a porous film for humidity sensing. The experimental results presented in this study demonstrate the reflective spectrum characteristics of the fabricated temperature and humidity sensor.

It can be published once the following issues are addressed.

 Response: We thank the Reviewer’s valuable and important comments on our manuscript. According to the Reviewer’s comments and kind suggestions, we have revised the manuscript. We have promoted the language of the manuscript.

  1. As shown in figure 4, what is the effect of parametric variations on the spectrum?

Response: Thanks for the comment. For the FBG, if the grating period is change, then the Bragg wavelength will change i.e. the FBG peak will shift. For the thin film, if the thin film thickness or the effective refractive index is change, the free spectral region of the thin film interference will change i.e. the troughs will shift.

  1. What do you think of this sensor's performance in the O-band?

Response: Thanks for the comment. In my opinion, if the grating period and thin film thickness are adjusted to match the O-band, the performance of the new sensor will similar to this manuscript, but the temperature and humidity sensitivity will have slight changes. And the new sensor will match with the optical instruments in the O-band.

  1. Why was the wavelength range of 1520 nm to 1620 nm chosen for the infrared light source? Is this range relevant to the sensor's application?

Response: Thank you for the comment. Yes, the range of 1520 nm to 1620 nm is more suitable to the optical fiber application. At present, the commonly used FBG is 1550 nm, and communication fiber is designed to have the best performance at 1550 nm. Therefore, the work wavelength of most optical fiber devices are in the C-band.

  1. How was the temperature experiment carried out? Were the temperature conditions accurately controlled and stable during the experiment?

Response: Thanks for the suggestion. The temperature resolution of the blast drying oven is 0.1°C. The relevant content is revised as follows.

During the temperature experiment, in order to stabilize the interference spectrum of the sensor, the sensor is put into a stable humidity environment which is created with a saturated potassium chloride solution. Subsequently, The airtight bottle (sensor probe and saturated salt solution are within it) is put into the blast drying oven (9055A, Shanzhi instrument equipment Co., LTD) to conduct the temperature test. When the temperature setting value is reached, the environment temperature is maintained for 10 minutes.

  1. Can you elaborate on why the porous film was placed in a stable humidity environment during the temperature experiment? How does humidity stability affect temperature measurements?

Response: Thank you for the comment. Putting the sensor probe into a stable humidity environment is to ensure that the interference spectrum of the porous thin film is stable. If the humidity is not stable (in other words, the interference spectrum is not stable), the temperature measurements can not be affected on account of the FBG is insensitive to humidity.

  1. As shown in Fig. 5, are the reported Bragg wavelength shifts consistent with theoretical expectations for the given temperature range?

Response: Thanks for the comment. Yes, the temperature sensitivity is consistent with the theoretical expectation. The temperature sensitivity of the sensor is 10.4 pm/°C, and the theoretical expectation in the part of temperature sensing theory is 0.0109 nm/°C (i.e. 10.9 pm/°C), therefore the experimental result is consistent with the theoretical expectation. The temperature sensitivity of ordinary FBG is around 10 pm/°C.

  1. How confident are you in the accuracy of the 13 nm wavelength shift observed in the characteristic valley? What factors contribute to measurement variability?

Response: Thanks for the comment. We are pretty sure about the accuracy of the 13 nm wavelength shift. The humidity measurement is affected by the thin film thickness and its porosity. In generally, thinner film or higher porosity is good to improve the humidity sensitivity, but it will also decrease the robustness of the humidity sensitive film.

  1. Were there any comparative analyses of existing sensors or benchmarks in terms of accuracy, sensitivity, or other performance metrics? Please provide a table for comparisons.

Response: Thanks for the suggestion. We have added the comparative analyses which is as follows.

The humidity sensing performance of the sensor compared with previous publications is shown in Table 1. From Table 1, it can be seen that the ceramic materials can be used as the humidity sensitive film of ultra-low humidity detection. The anodized alumina film is easy to control its porosity and thickness, which make it suitable for the ultra-low humidity detection. In ultra-low humidity environment, the humidity detection is strongly influenced by temperature. Thus, the sensor configuration within this manuscript will contribute to investigate the relationship between humidity and temperature at ultra-low humidity environment.

Table 1. Performance comparison of the reported work with previous publication.

Prior art

Material

Measuring range (%RH)

sensitivity

[9]

Chitosan

40 - 92

7.15 nm/% RH

[10]

Agarose

10 - 95

0.0024 nm·%−1

[11]

Nafion

30 - 85

3.78 nm/%RH

[12]

Polyvinyl alcohol

>70

−0.4573 dB/%RH

[13]

Polyimide

23.8 - 83.4

1.832 pm/%RH

[14]

SiO2

0.06 - 70

0.3 nm/%RH

[15]

SnO2

20 - 90

1.9 nm/%RH

[16]

ZnO

20 - 95

25.2 mV/%RH

[17]

TiO2

1.8 - 74.7

0.43 nm/%RH

[18]

Al2O3

3 – 98 (and 180 - 1000 ppm)

/

Reported work

Al2O3

11.3 – 84.3

185 pm/%RH

Reviewer 4 Report

Reviewer’s comments:

1.     In the manuscript, the literatures should be citated for the formulas.

2.     The image of FBG temperature sensor should be exhibited in the manuscript also.

3.     The latest literatures should be also referred from 2023.

4.     The research data of sensor in sensitivity should compare with other reports for revealing the advanced research.

The style and format of native language in English should be confirmed in the content.

Author Response

Response: We thank the Reviewer’s valuable and important comments on our manuscript. According to the Reviewer’s comments and kind suggestions, we have revised the manuscript. We have promoted the language of the manuscript.

  1. In the manuscript, the literatures should be citated for the formulas.

Response: Thanks for the suggestion. The literatures is cited for the formulas. The relevant content is revised as follows.

The Bragg wavelength can be calculated by formula 1 [26].

The Bruggeman medium model is an effective method to calculate the effective refractive index, which is expressed as follows [27].

The total reflective light intensity of the dual beam interference can be expressed as formula 8 [28].

  1. The image of FBG temperature sensor should be exhibited in the manuscript also.

Response: Thanks for the suggestion. About the Figure 2: temperature and humidity dual sensor structure, as shown in the following, the part (1) is the FBG which is used to detect temperature, and the part (2) is the humidity sensitive film which is used to detect humidity.

  1. The latest literatures should be also referred from 2023.

Response: Thank you for the suggestion. We have added the literatures of 2023 year.

  1. The research data of sensor in sensitivity should compare with other reports for revealing the advanced research.

Response: Thanks for the suggestion. We have added the comparative analyses which is as follows.

The humidity sensing performance of the sensor compared with previous publications is shown in Table 1. From Table 1, it can be seen that the ceramic materials can be used as the humidity sensitive film of ultra-low humidity detection. The anodized alumina film is easy to control its porosity and thickness, which make it suitable for the ultra-low humidity detection. In ultra-low humidity environment, the humidity detection is strongly influenced by temperature. Thus, the sensor configuration within this manuscript will contribute to investigate the relationship between humidity and temperature at ultra-low humidity environment.

Table 1. Performance comparison of the reported work with previous publication.

Prior art

Material

Measuring range (%RH)

sensitivity

[9]

Chitosan

40 - 92

7.15 nm/% RH

[10]

Agarose

10 - 95

0.0024 nm·%−1

[11]

Nafion

30 - 85

3.78 nm/%RH

[12]

Polyvinyl alcohol

>70

−0.4573 dB/%RH

[13]

Polyimide

23.8 - 83.4

1.832 pm/%RH

[14]

SiO2

0.06 - 70

0.3 nm/%RH

[15]

SnO2

20 - 90

1.9 nm/%RH

[16]

ZnO

20 - 95

25.2 mV/%RH

[17]

TiO2

1.8 - 74.7

0.43 nm/%RH

[18]

Al2O3

3 – 98 (and 180 - 1000 ppm)

/

Reported work

Al2O3

11.3 – 84.3

185 pm/%RH

Round 2

Reviewer 1 Report

The authors have addressed all of my questions. 

Reviewer 2 Report

The response and revised manuscript are satisfactory, and it is recommended to accept.

Reviewer 3 Report

Thanks for the response. No further comments.